# Evaluating Graph Generative Models with Graph Kernels: What Structural Characteristics Are Captured?

**Martijn Gösgens**[*]                                              *research@martijngosgens.nl*

**Alexey Tikhonov**                                                    *altsoph@gmail.com*
*Independent researcher*

**Liudmila Prokhorenkova**                                    *ostroumova-la@yandex-team.com*
*Yandex Research*

**Reviewed on OpenReview:** *https://openreview.net/forum?id=d9MhajJTO4*

## Abstract

For many practical problems, it is important to measure similarity between graphs. This can be done via *graph kernels*. One particular application where the choice of a graph kernel is essential is assessing the quality of graph generative models. However, despite the vast number of graph kernels available in the literature, only basic kernels are usually considered for generative model evaluation. In this paper, we fill this gap and analyze how different graph kernels perform as an ingredient in the pipeline of generative model performance evaluation. To conduct a detailed analysis, we propose a framework for comparing graph kernels in terms of which high-level structural properties they are sensitive to: heterogeneity of degree distribution, the presence of community structure, the presence of latent geometry, and others. For this, we design continuous transitions between random graph models that affect a particular property and measure which graph kernel is sensitive to the corresponding change. We show that using such diverse models with the corresponding transitions is crucial for evaluation: many kernels can successfully capture some properties and fail on others. We also found some well-known kernels that show good performance in our experiments but have been previously overlooked in the literature on evaluating graph generative models.

## 1 Introduction

Many real-world objects can be represented as graphs: social and citation networks, molecules, the Internet, transportation networks, and so on. For a number of practical problems, it is important to measure similarity or distance between graphs, e.g., graph classification, graph clustering, or evaluating graph generative models.

A number of *graph kernels* have been proposed in the literature to evaluate the similarity between graphs (Kriege et al., 2020; Nikolentzos et al., 2021). Graph kernels are often based on some graph statistics like node degrees, shortest path lengths, small subgraph counts, and so on. Thus, different kernels capture different graph properties. However, it is not obvious *which* properties are captured by a given kernel. This makes it challenging to choose a suitable kernel for a particular problem.

One important application of graph kernels is evaluating the performance of graph generative models. To evaluate such a model, one needs a measure that compares a generated set of graphs with a reference set, which is a non-trivial task (O'Bray et al., 2022; Thompson et al., 2022). A standard approach is to convert all graphs to some vector representations or to compute similarities (kernel values) for pairs of graphs and then aggregate the values into a similarity measure. The most widespread measure of (dis)similarity is maximum mean discrepancy (MMD) which measures the distance between two sets of elements given the

---

[*]This research was conducted while Martijn Gösgens was employed by Eindhoven University of Technology.

pairwise kernel values. The problem of choosing the right measure for graph generative model evaluation has gained significant attention recently (O'Bray et al., 2022; Thompson et al., 2022; Shirzad et al., 2022). In particular, it was shown that the choice of a kernel (or even a particular parameter for a given kernel) for MMD computation may drastically affect the outcome of the model comparison (O'Bray et al., 2022).

Despite recent research, the problem of choosing the right graph kernel for evaluating graph generative models is under-explored. Indeed, despite the vast number of graph kernels available in the literature, only very basic options are usually considered for generative model evaluation, e.g., based on the degree distribution or clustering coefficient. In this paper, we fill this gap and analyze how different graph kernels perform as an ingredient of the generative model performance evaluation pipeline. To conduct a detailed analysis, we propose a framework for comparing graph kernels in terms of their sensitivities to different graph properties. For this, we choose several pairs of random graph models that are different in a particular graph property — the presence of community structure, degree distribution, latent geometry, etc. Then, we design continuous transitions between the models: in each transition, the presence of a given property gradually increases. To evaluate whether a particular kernel is sensitive to a particular type of change, we generate several sets of graphs at different points of the transition and check whether the kernel is able to distinguish these different sets. As a result, we provide a detailed comparison of graph kernels that can help practitioners in choosing the best one for their application. Our framework aligns well with how the performance of graph generative models is evaluated and thus can help in the development of better measures for this task.

One of our main observations is that many kernels are sensitive to some of the properties while being insensitive to others. For example, our experiments show that the popular Weisfeiler-Leman kernel is insensitive to geometry, which makes it unsuitable for tasks like molecular modeling, where the nodes (atoms) have spatial locations. This shows the importance of using different structural properties for assessing which characteristics a kernel is sensitive to. Among the best-performing kernels in our experiments is the long-known Shortest Path kernel that has not been used previously in the context of evaluating the performance of graph generative models. Other good-performing kernels are the Graphlet kernel, the Pyramid Match kernel, and the kernel based on the NetLSD-wave graph representation.

The proposed framework and the obtained results can help practitioners in choosing suitable kernels for graph generative model analysis.

## 2 Preliminaries

In this work, we analyze graph kernels and focus on undirected graphs $G = (V, E)$ with no node/edge attributes since most of the known kernels can be applied to such graphs.

### 2.1 Measuring graph similarity

Various approaches can be used to measure similarity or dissimilarity between graphs. Two major research directions are concentrated on *graph distances* and *graph kernels*.

*Graph distances* measure dissimilarity between graphs and are supposed to satisfy the metric axioms. However, the positivity axiom is usually violated in the sense that the distance between two different graphs can be equal to zero. Indeed, if we guarantee that $D(G, G') = 0$ if and only if $G$ and $G'$ are isomorphic, then computing such distance is at least as hard as graph isomorphism testing, which is infeasible for most applications.

In turn, a *graph kernel* is a symmetric, positive semidefinite function defined on the space of graphs. This function can be expressed as an inner product in some Hilbert space. A survey of many graph kernels can be found in, e.g., Kriege et al. (2020); Nikolentzos et al. (2021).

We conduct a comparative analysis of graph kernels. Let us note that any kernel $K(\cdot, \cdot)$ can be transformed to a distance measure (up to the positivity axiom) as, e.g., $D(G_1, G_2) = \sqrt{K(G_1, G_1) + K(G_2, G_2) - 2K(G_1, G_2)}$. Similarly, there are multiple ways of transforming a graph distance to a graph kernel.

A typical approach to defining a kernel or distance for graphs is to describe each graph based on its characteristics: degree distribution, substructure counts, shortest path lengths, spectral properties, and so on. The obtained descriptors can then be used to compute a graph kernel or graph distance. Some kernels are defined for graphs with discrete node labels. However, they can be applied to unlabeled graphs as well: for this, we assume that all node labels are equal.

Let us now define several known graph kernels.

**Degree histogram (DH) kernel**  A graph $G$ is represented by a vector $f_G$, where the $i$-th coordinate is the number of nodes of degree $i$. Then, the degree histogram kernel is computed as the scalar product of $f_G$ and $f_{G'}$. Equivalently, the degree histogram kernel is given by

$$\mathrm{DH}(G_1, G_2) = \sum_{u \in V(G_1)} \sum_{v \in V(G_2)} \mathbb{1}\{\deg(u) = \deg(v)\},$$

where $\deg(u)$ be the degree of $u$.

**Weisfeiler-Leman (WL) subtree kernel**  was proposed in Shervashidze et al. (2011) and is based on the WL color refinement procedure. This procedure works as follows. Initially, all nodes have their labels (or one fixed label for unlabelled graphs). At each iteration, a node's label is replaced by another label identifying a multiset of labels of its neighbors. The procedure stops when it converges. Based on that, the WL kernel for $l$ iterations is

$$\mathrm{WL}(G_1, G_2) = \sum_{i=0}^{l} \sum_{u \in V(G_1)} \sum_{v \in V(G_2)} \mathbb{1}\{\mathrm{wl}_{G_1}^i(u) = \mathrm{wl}_{G_2}^i(v)\},$$

where $\mathrm{wl}_{G_1}^i(u)$ is the label at $i$-th iteration of the WL procedure.

**The Shortest Path (SP) kernel**  was introduced in Borgwardt & Kriegel (2005). Each shortest path is described by the following triplet: its length and labels of starting and ending nodes. Then, the graph is represented by a vector $f_G$, where each coordinate is the frequency of a particular triplet in a graph. Then, the kernel is a scalar product of $f_G$ and $f_{G'}$. In our setting, where each node has the same label, the SP kernel compares the shortest-path histograms. Thus, the vector $f_G$ can be interpreted as a signature of the topology described by the network.

**The Graphlet kernel**  was proposed in Pržulj (2007). *Graphlets* are small connected subgraphs of a graph. Consider graphlets of size $k$. We assign an index to each graphlet and let $f_G$ be a vector such that its $i$-th entry is equal to the frequency of occurrence of $i$-th graphlet in $G$. Then, the kernel can be computed as the scalar product of $f_G$ and $f_{G'}$.

**Weisfeiler-Lehman optimal assignment (WL-OA) kernel**  was introduced by Kriege et al. (2016) and it improves the WL kernel by finding the optimal matching of nodes. Formally,

$$\mathrm{WL\text{-}OA}(G_1, G_2) = \max_{B \in \mathcal{B}(V_1, V_2)} \sum_{v_1, v_2 \in B} k(v_1, v_2),$$

where $\mathcal{B}(V_1, V_2)$ is the set of all bijections between the node sets and

$$k(v_1, v_2) = \sum_{i=0}^{l} \mathbb{1}\{\mathrm{wl}_{G_1}^i(v_1) = \mathrm{wl}_{G_2}^i(v_2)\}.$$

Since WL-OA requires a bijections between the nodes, it can be applied only to graphs of the same size.

**The Neighborhood Subgraph Pairwise Distance (NSPDK) kernel** proposed in Costa & De Grave (2010) considers pairs of rooted subgraphs of radius $r' \leq r$ whose roots are located at distance $d' \leq d$ from each other. A kernel $k_{r',d'}(G_1, G_2)$ counts the number of such pairs of rooted subgraphs in the first graph that are identical to pairs in the second graph. Then,

$$\text{NSPD}(G_1, G_2) = \sum_{r'=0}^{r} \sum_{d'=0}^{d} \frac{k_{r',d'}(G_1, G_2)}{k_{r',d'}(G_1, G_1) k_{r',d'}(G_2, G_2)}.$$

To make the computation of this kernel feasible, graph invariants can be employed to encode each rooted subgraph. Then, these invariants can be compared instead of graph isomorphism checking.

**NetLSD** treats a graph as a dynamical system and simulates heat and wave diffusion processes on nodes and edges of a given graph, followed by measuring system conditions at fixed timestamps (Tsitsulin et al., 2018). More formally, let $\lambda_j$ be the $j$-th smallest eigenvalue of the normalized Laplacian of a graph $G$. For a timestamp $t$, we define the *heat trace* $h_t$ and *wave trace* $w_t$ of a graph $G$ as follows:

$$h_t = \sum_j e^{-t\lambda_j}, \quad w_t = \sum_j e^{-it\lambda_j}. \tag{1}$$

Here $t > 0$ for the heat trace and $t \in [0, 2\pi)$ for the wave trace.

Then, the *heat trace signature* and *wave trace signature* of $G$ are defined as the sequences of the corresponding traces at different timestamps, i.e., $h(G) = \{h_t\}_{t \in \mathcal{T}_h}$ and $w(G) = \{w_t\}_{t \in \mathcal{T}_w}$. As in the original article, we use 250 log-spaced time stamps between $10^{-2}$ and $10^2$ for $\mathcal{T}_h$ and 250 equally-spaced time stamps between 0 and $2\pi$ for $\mathcal{T}_w$, respectively.

Finally, the NetLSD distance (heat or wave) between two graphs $G$ and $G'$ can be computed as any distance measure between the corresponding signatures. In our comparison, we use NetLSD representations to obtain a graph kernel. For this, we measure the cosine similarity between the graph representations.

**Pyramid Match kernel** proposed in Nikolentzos et al. (2017) first embeds the vertices of each graph into a low-dimensional vector space using the eigenvectors of the $d$ largest (in magnitude) eigenvalues of the graph's adjacency matrix. Since the signs of these eigenvectors are arbitrary, it replaces all their components by their absolute values. Each vertex is thus a point in the $d$-dimensional unit hypercube. To find an approximate correspondence between the sets of vertices of two graphs, the kernel maps these points to multi-resolution histograms, and compares the emerging histograms with a weighted histogram intersection function, see Nikolentzos et al. (2017) for more details.

## 2.2 Kernels for evaluating graph generative models

To evaluate a graph generative model, one needs to compare a set of graphs produced by the model with some reference set of graphs (not available to the model during training). Typically, the comparison consists of two steps. First, each graph is described as a point in a vector space, or for each pair of graphs their similarity is defined. Then, two sets of points or two sets of similarity values are compared. Arguably the most widely-used approach is to define a kernel $K(\cdot, \cdot)$ for measuring the similarity of graphs and then use *maximum mean discrepancy* (MMD) to measure the distance between two sets of graphs $\mathcal{G}_1$ and $\mathcal{G}_2$:

$$\text{MMD}^2(\mathcal{G}_1, \mathcal{G}_2) = \frac{1}{|\mathcal{G}_1|^2} \sum_{G_1, G_2 \in \mathcal{G}_1} K(G_1, G_2) + \frac{1}{|\mathcal{G}_2|^2} \sum_{G_1, G_2 \in \mathcal{G}_2} K(G_1, G_2) - \frac{2}{|\mathcal{G}_1||\mathcal{G}_2|} \sum_{G_1 \in \mathcal{G}_1} \sum_{G_2 \in \mathcal{G}_2} K(G_1, G_2).$$

If $K(G_1, G_2)$ is a kernel, then it can be expressed as $K(G_1, G_2) = \langle f(G_1), f(G_2) \rangle$ for some vector representation $f(G)$. Then, $\text{MMD}^2 = \|\mu_1 - \mu_2\|^2$, where $\mu_i = \frac{1}{|\mathcal{G}_i|} \sum_{G \in \mathcal{G}_i} f(G)$. Thus, MMD is indeed a distance. Besides MMD, there are other approaches to compare distributions: Fréchet Distance (Heusel et al., 2017), Improved Precision & Recall (Kynkäänniemi et al., 2019), Density & Coverage (Naeem et al., 2020). Overall, there are many ways to compare two distributions of graphs.

The work by O'Bray et al. (2022) is the most relevant to our study. The authors consider graphs without node features and labels and compare several typically used MMD-based measures. To define a kernel, O'Bray

et al. (2022) describe each graph by a vector via simple structural characteristics: a histogram of the degree distribution, a histogram of local clustering coefficients, or eigenvalues of the Laplacian. Then, different transformations for obtaining a kernel are considered: the first Wasserstein distance (EMD), total variation distance (TV), and the radial basis function kernel (RBF). Each transformation has a parameter that needs to be specified. The main conclusion of their paper is that for different kernels and their hyperparameters, the outcome of the models' comparison may drastically differ. In the present paper, we note that the two-step procedure of first computing a graph representation and then applying a transformation to get a kernel can be replaced by considering any of the kernels available in the literature such as the shortest path or the graphlet kernels described above. Thus, we cover various graph kernels and compare their performance.

Another contribution of O'Bray et al. (2022) is their approach to comparing measures: they propose to make perturbations to a given set of graphs and measure whether MMD correlates with the degree of change. The considered modifications are the following: random edge insertions, random edge deletions, random rewiring operations, and random node additions. The key distinction of our paper is that we design specific graph distribution changes that target particular graph properties and allow us to give a much more detailed understanding of what can be captured by a particular kernel.

Thompson et al. (2022) also compares evaluation measures for graph generative models and assumes that graphs may have features of nodes and edges. They consider two types of measures: classic (as in O'Bray et al. (2022)) and based on neural representations. Thompson et al. (2022) suggest adopting the widely used measures from image generation literature (Fréchet Distance, Improved Precision & Recall, Density & Coverage). But for graphs (in contrast to images), this would require training a neural network for each dataset. Instead, the authors suggest using representations obtained via a randomly initialized Graph Isomorphism Network (GIN) (Xu et al., 2019). To obtain a graph representation, a readout function is applied to aggregate node representations at each GIN layer and then the obtained vectors are concatenated. In the experiments, Thompson et al. (2022) measure the rank correlation between the degree of perturbation and the measure and show that the proposed randomly initialized GNNs work well. Several aspects are tested: fidelity (whether a measure can detect random graphs added to a set of graphs or detect randomly rewired edges); diversity (a measure should be sensitive to mode dropping and mode collapse); sensitivity to node and edge features; sample efficiency (the minimum number of samples to discriminate a set of random graphs from real samples). However, in terms of graph structure, only random rewiring and the Erdős–Rényi random graph model are considered. Finally, Shirzad et al. (2022) propose replacing a randomly initialized GIN model with a contrastively trained GNN model, which makes the obtained evaluation measure dataset-dependent.

## 3 Comparing graph kernels

In this section, we describe the proposed framework for comparing graph kernels. The main questions we want to address is 'What structural characteristics are captured by what kernel?'. With 'captured', we mean that the kernel can distinguish two sets of graphs if they differ in that particular aspect. To measure the sensitivity of a kernel to a graph characteristic, we consider a sequence of graph generators where, at each step, the considered graph characteristic is more present. To that end, we consider graph generators that include a *step* parameter $\theta$ which we vary between 0 and 1 to increase the value of the characteristic. We describe below which graph generators we consider and how we interpolate between them.

In all of our generators, we keep the number of nodes $n$ constant, and we also preserve the expected number of edges $m$ (except when the target characteristic is density). We use the Erdős-Rényi (ER) random graph model as *baseline* generator. In the Erdős-Rényi model, each edge is added independently with probability $p = m / \binom{n}{2}$.

As we discussed above, previous studies (O'Bray et al., 2022; Thompson et al., 2022) measured the sensitivity of graph kernels by perturbing a given (real) dataset. Instead, we use random graph generators since they allow us to gradually increase or decrease specific characteristics and thus conduct a detailed analysis of kernels' sensitivity to various structural shifts.

### 3.1 Structural graph characteristics

In this section, we discuss structural graph characteristics that we consider in our research and introduce the generative models that are used for each of these characteristics.

**Density**   Density is the simplest graph characteristic. We model this by varying the probability $p$ in the basic Erdős-Rényi model.

**Degree heterogeneity**   The degree distribution of an ER graph differs from what is observed in real-world networks: ER graphs have Binomially distributed degrees, leading to a variance that is lower than the mean degree. However, many real-world networks have a much more *heterogeneous* degree distribution, where the variance is often many times larger than the mean degree. Many networks even seem to have *power-law* degree distributions, leading to many hubs and a high degree variance (Barabási & Bonabeau, 2003). Several generative models incorporate degree heterogeneity by prescribing an (expected) degree sequence, as can be done with the Configuration Model and the Chung-Lu model (Chung & Lu, 2002; Van der Hofstad, 2016). We use the Chung-Lu model because it is a generalization of ER and is simple to work with. The input to this model is the vector of the expected degrees $(w_1, \ldots, w_n)^T$. Given the expected degrees, an edge between two nodes $u$ and $v$ is added with probability $\frac{w_u w_v}{\sum_i w_i}$ independently of all other edges. We sample the prescribed degrees from a Pareto distribution with power-law exponent $\gamma \geq 2$ and scale parameter chosen to ensure that the expected number of edges equals $m$. In the limit $\gamma \to \infty$, this is equivalent to the ER model, while small finite values lead to degree sequences with high variance.

**Clustering**   In many real-world networks, nodes with common neighbors are more likely to be connected to each other (Watts & Strogatz, 1998; Van der Hofstad, 2016). This phenomenon is often referred to as *clustering* (Holland & Leinhardt, 1971; Peixoto, 2022), and it results in an abundance of triangles, which is often quantified by the *clustering coefficient*. Social networks and many other real graphs are well-known for having a high degree of clustering (Holland & Leinhardt, 1971; McPherson et al., 2001).

There are two main mechanisms that are used to explain and model clustering: *community structure* and *latent geometry*. In our analysis, we treat these as two different graph characteristics.

**Community structure**   Many real-world networks contain groups of nodes that are more densely connected to each other than to the rest of the network. In network science, these groups are referred to as *communities* (Fortunato, 2010), and they often have a natural interpretation, like friend groups in social networks or subject areas in citation networks. In addition, the presence of community structure can explain the clustering that is observed in real networks: the presence of densely connected groups leads to an increased number of triangles.

The simplest generative model for community structure is the *Planted Partition* (PP) model (Holland et al., 1983), where we are given a partition of the network nodes into communities, and node pairs of the same community connect with probability $p_{\text{in}}$, while node pairs of different communities connect with probability $p_{out}$. We consider two communities of size $n/2$ each, and we parameterize $p_{\text{in}}, p_{out}$ as

$$p_{\text{in}}(\lambda) = \frac{4m\lambda}{n^2(1+\lambda) - 2\lambda n}, \ \ p_{out}(\lambda) = \frac{4m}{n^2(1+\lambda) - 2\lambda n}, \tag{2}$$

so that the expected number of edges equals $m$, while $p_{\text{in}}/p_{out} = \lambda$. This parametrization of the PP model reduces to the ER model for $\lambda = 1$, and the (expected) global clustering coefficient increases monotonously with $\lambda$.

**Latent geometry**   Community structure explains clustering by assuming a certain (finite) set of types, and that nodes of the same type have a higher likelihood of connecting than nodes of different types. The *Random Geometric* (RG) model generalizes this notion by considering a continuous type space and assuming that nodes whose types are *similar* to each other have a larger probability of connecting to each other. In citation networks, for example, papers tend to cite papers on related topics. This continuous type space can

be thought of as some *feature space*, and the (dis)similarity may be quantified by some distance measure in this space. However, often one only has access to the network connections, and not the positions in the feature space. In such cases, we say that the network has a *latent geometry*.

The simplest way to model this is by assigning to each node a coordinate in some *latent space*, and connecting two nodes if their distance is lower than some threshold. This model is referred to as a *random geometric graph* (Penrose, 2003). We consider a two-dimensional *torus* geometry, where each node is assigned a coordinate in $[0,1)^2$ uniformly at random, and two nodes are connected whenever their distance is below some threshold $r$. We choose $r$ to ensure that the network has $m$ edges in expectation, see Appendix A.1.

**Dimensionality** Whenever a network has latent geometry, its connections depend highly on the characteristics of that geometry. For example, networks with hyperbolic geometry tend to have a high level of degree heterogeneity (Krioukov et al., 2010). In particular, the dimension of the latent space heavily affects both the local properties (e.g., clustering) and global characteristics (e.g., diameter). In general, the clustering coefficient decreases with dimension.

To model varying dimensionality, we use the random geometric graph model: we consider a torus with width 1 and height $h \in (0,1]$. For $h = 1$, this corresponds to the standard two-dimensional torus, while the limit $h \downarrow 0$ results in a one-dimensional torus, i.e., a circle. Note that the radius $r$ that leads to $m$ edges in expectation, depends on $h$. The desired value of $r(h,m)$ is derived in Appendix A.1.

**Complementarity** In some particular types of networks, such as protein-protein interaction networks and economic networks (Talaga & Nowak, 2022; Mattsson et al., 2021), it has been observed that the clustering coefficient is significantly lower than that of an ER graph with the same number of edges. Instead, these graphs have a large number of quadrangles (cycles of length four), leading to a locally bipartite structure (Estrada, 2006). This phenomenon is usually explained by *complementarity* (Talaga & Nowak, 2022): in these networks, nodes do not connect if they are similar to each other, but if they differ in some specific way. In economic networks, for example, companies that produce a certain product will trade with companies that are in need of that particular product. This leads 'similar' nodes to have many common neighbors, but rarely a direct connection. In turn, dissimilar nodes have a high likelihood of connecting.

There are several random graph models for complementarity: one option is *disassortative* community structure, e.g., by a PP model with $p_{\text{in}} < p_{\text{out}}$, but there are also ways to model complementarity using latent geometry. We model complementarity by a latent spherical geometry (Talaga & Nowak, 2022). This has the nice property that each point in the latent space has a unique *antipodal* point at maximum distance. We assign each node to a point on the hypersphere (with radius 1) uniformly at random and connect two nodes if their distance exceeds $\pi - r$ for some value of $r$, chosen to ensure $m$ edges in expectation. We refer to this generator as the *Spherical Complementarity* (SC) model.

## 3.2 Interpolating graph characteristics

For each graph characteristic, we define an interpolation between two graph generators, so that the strength of that characteristic changes monotonously along the transition. We parametrize each of these interpolations by a step parameter $\theta \in [0,1]$, so that $\theta = 0$ leads to the 'left' generator, while $\theta = 1$ leads to the 'right' generator. We make use of the fact that most of the generating models introduced in Section 3.1 are generalizations of the ER model. In these cases, we simply parameterize a transition away from the ER model. The ER model is chosen with $p = m/\binom{n}{2}$ so that the expected number of edges equals $m$. For the geometric models used for latent geometry and complementarity, we will take a different approach to interpolate between generators.

**Density (ER($p$))** We use the ER model with the edge probability $p(\theta + 1)$. Thus, when $\theta$ changes from 0 to 1, the edge probability increases from $p$ to $2p$.

**Heterogeneity (ER$\leftrightarrow$CL)** We use the Chung-Lu (CL) model with weights drawn from a Pareto distribution with power-law exponent $1 + 1/\theta$ and scale chosen such that the expected number of edges equals $m$. Note that $\theta = 0$ leads to a power-law exponent $\infty$, i.e., all the weights are constant and the corresponding

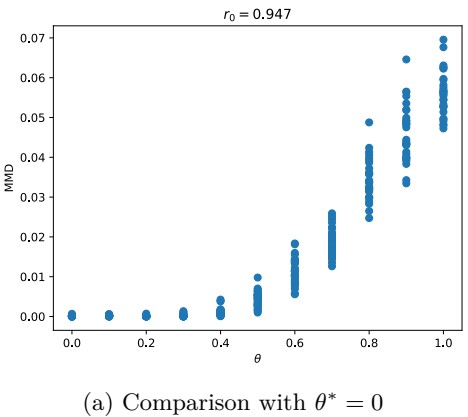 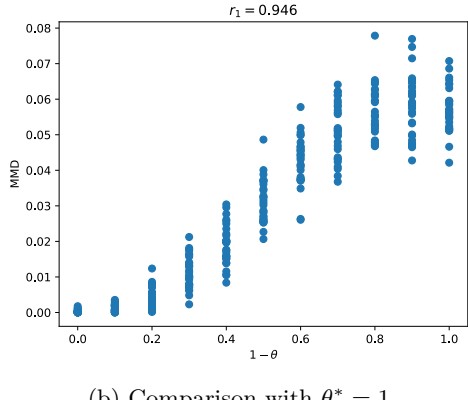

(a) Comparison with $\theta^* = 0$                               (b) Comparison with $\theta^* = 1$

Figure 1: Scatter plots of the MMD values of the SP kernel for the ER↔CL interpolation.

CL model is equivalent to the ER model. The values $\theta > 0.5$ give weight distributions with infinite variance, which leads to a high variance in the degree distribution.

**Communities (ER↔PP)** We use the PP model with $p_{in}$, $p_{out}$ as given in equation 2 for $\lambda = 1 + c(n,m)\theta$, where $c(n,m)$ is a constant so that $\theta = 0$ leads to $p_{in} = p_{out} = p$, while $\theta = 1$ leads to every vertex having (in expectation) one neighbor outside its community. This value is derived in Appendix A.2.

**Latent geometry (ER↔Torus)** For this transition, we take the *mixture* of an ER graph and a RG torus graph: we generate a graph from each generator and denote their adjacency matrices by $A^{(ER)}$ and $A^{(RG)}$. We construct the adjacency matrix $A$ as follows: for each node pair with indices $1 \leq i < j \leq n$ we draw an independent Bernoulli random variable $B_{ij}$ with success probability $\theta$ and set $A_{ij} = A_{ij}^{(ER)} + B_{ij}(A_{ij}^{(RG)} - A_{ij}^{(ER)})$. Hence, $\theta = 0$ leads to $A = A^{(ER)}$, while $\theta = 1$ leads to $A = A^{(RG)}$. We complete the adjacency matrix symmetrically: $A_{ij} = A_{ji}$ for $i > j$ and $A_{ii} = 0$.

**Dimensionality (Torus↔Circle)** To interpolate between a two- and one-dimensional torus, we consider a torus of height $h = 1 - \theta$ and connection radius $r(h,m)$ as derived in Appendix A.1. This way, $\theta = 0$ leads to a torus on $[0,1)^2$ while $h \downarrow 0$ leads to a one-dimensional torus (a circle).

**Complementarity (ER↔SC)** For this transition, we take the same approach as for the interpolation ER↔Torus: we construct a graph by taking a mixture of graphs generated by the ER and SC models, such that $\theta = 0$ results in an ER graph, while $\theta = 1$ results in a SC graph.

### 3.3 Measuring the sensitivity of a kernel

We now explain how we quantify the sensitivity of a kernel w.r.t. an interpolation between graph generators. We consider a discretization $\Theta \subset [0,1]$ of the interpolation such that $\{0,1\} \subset \Theta$. For $\theta \in \Theta$, let $\mathcal{G}_\theta$ denote a set of $g$ graphs sampled independently from the interpolation generator at step $\theta$. Furthermore, let $\text{MMD}(\mathcal{G}_{\theta_1}, \mathcal{G}_{\theta_2}; K)$ denote the MMD value between $\mathcal{G}_{\theta_1}$ and $\mathcal{G}_{\theta_2}$ w.r.t. the kernel $K$. We compare each $\theta \in \Theta$ to each endpoint $\theta^* \in \{0,1\}$. For each endpoint $\theta^* \in \{0,1\}$, we quantify the sensitivity of the kernel $K$ w.r.t. the interpolation as the *Spearman correlation* between $\text{MMD}(\mathcal{G}_{\theta^*}, \mathcal{G}_\theta; K)$ and $|\theta - \theta^*|$. If this value is close to 1, it means that the MMD values tend to increase when transitioning $\theta$ away from $\theta^*$. But if this value is close to zero, it indicates that there is no clear monotone relation between $\theta$ and $\text{MMD}(\mathcal{G}_{\theta_1}, \mathcal{G}_{\theta_2}; K)$. This leads to two different correlation coefficients $r_0$ and $r_1$, corresponding to the different values of $\theta^*$. See Figure 1 for an illustration.

To ensure the independence of the different MMD values that are used to compute a correlation coefficient, we sample two different sets of graphs for each MMD value. Between each pair of $\theta$'s that we compare, we

compute $s$ MMD values. This results in sampling $s \cdot g$ graphs for each $\theta \in \Theta \setminus \{0, 1\}$, and $(1 + |\Theta|) \cdot s \cdot g$ graphs for each $\theta^* \in \{0, 1\}$.

## 4 Experiments

### 4.1 Setup

We follow the framework described in the previous section. In most of our experiments, we consider graphs with $n = 50$ nodes and (in expectation) $m = 190$ edges. We discretize the interpolation interval $[0, 1]$ by $\Theta = \{0.0, 0.1, \ldots, 1.0\}$. Thus, we have $|\Theta| = 11$ steps in our interpolation. We consider sets of $g = 100$ graphs and compute $s = 30$ different MMD values for each pair of interpolation steps that we compare.

In this work, we mostly focus on small graphs since evaluating graph generative models is typically applied to small networks: e.g., in molecular modeling, the graphs consist of a relatively small number of atoms. Additionally, let us note that not all kernels are scalable: for instance, the graphlet kernel has a prohibitively high computational cost. Thus, for smaller graphs, we can include all popular graph kernels in our experiments. However, for the completeness of the study, we also consider larger graphs with $n = 1000$ nodes and (in expectation) $m = 3800$ edges, so that the average degree is the same in both experiments.

In our experiments, we use the following kernels and representations:

- Node degree histogram kernel (Degree);
- Shortest Path (SP) kernel (Borgwardt & Kriegel, 2005);
- Weisfeiler-Leman (WL) kernel (Shervashidze et al., 2011) with $l = 5$;
- Weisfeiler-Lehman optimal assignment (WL-OA) kernel (Kriege et al., 2016) with $l = 5$;
- Graphlet kernel (Pržulj, 2007) with $k = 3$ and $k = 4$ (referred to as Graphlet-3 and Graphlet-4, respectively);
- Neighborhood Subgraph Pairwise Distance kernel (NSPDK) (Costa & De Grave, 2010) with $r = 3$, $d = 4$;
- Pyramid Match (PM) kernel (Nikolentzos et al., 2017) with Pyramid histogram level 4 and the dimension of the hypercube 6;
- NetLSD graph representations (Tsitsulin et al., 2018) with heat and wave diffusion process;
- Random GIN (RandGIN) representations (Thompson et al., 2022).

These kernels are described in Section 2.1.

For most graph kernels, we use the GraKeL python library (Siglidis et al., 2020). Following Nikolentzos et al. (2021), we normalize the kernel values as $K(G_1, G_2)/\sqrt{K(G_1, G_1)K(G_2, G_2)}$. Since our research compares graph kernels as ingredients of evaluation measures, we have to fix their hyperparameters before running the experiments. For the considered kernels, we use their default hyperparameters listed above (that are either the default parameters of the implementation or the most commonly used parameters in the literature). For the Graphlet kernel, however, the considered values of $k$ are limited by 4 due to high computational complexity of the kernel. For this kernel, we consider and compare its performance for two parameter values: $k = 3$ and $k = 4$. For NetLSD and RandGIN we use the implementation provided by the authors with the default parameters. In both cases, we use cosine similarity to convert graph representations to kernel values. Thus, for all the obtained kernels we have $K(G, G) = 1$. Our code and experiments are available at https://github.com/MartijnGosgens/graph-kernels.

### 4.2 Results and discussion

The results are shown in Table 1 and the respective computation times are reported in Appendix B. We see that most kernels perform well on some of the interpolations while performing badly on others. This shows the importance of using different interpolations for assessing the sensitivity of kernels. Recall that in this aspect, our framework extends the works by O'Bray et al. (2022); Thompson et al. (2022).

Table 1: Sensitivity of kernels to various structural characteristics measured via the Spearman correlation as described in Section 3.3. We measure two sensitivity values $r_0$ and $r_1$ for two endpoints and report their average $(r_0 + r_1)/2$. The top three results are colored.

| | Density ER($p$) | Heterogeneity (ER↔CL) | Communities (ER↔PP) | Geometry (ER↔Torus) | Dimensionality (Torus↔Circle) | Complementarity (ER↔SC) |
|---|---|---|---|---|---|---|
| Degree | **0.996** | **0.992** | 0.332 | 0.073 | 0.229 | 0.083 |
| SP | 0.994 | 0.953 | 0.959 | 0.957 | **0.985** | 0.929 |
| WL | **0.996** | **0.993** | 0.387 | 0.150 | 0.468 | 0.097 |
| WL-OA | **0.996** | **0.993** | 0.408 | 0.601 | 0.549 | 0.428 |
| Graphlet-3 | **0.996** | 0.988 | **0.979** | **0.972** | 0.721 | **0.972** |
| Graphlet-4 | **0.996** | 0.991 | **0.973** | **0.980** | **0.843** | **0.970** |
| NSPDK | 0.365 | 0.956 | 0.373 | 0.854 | 0.589 | 0.580 |
| PM | 0.981 | 0.990 | 0.950 | 0.922 | 0.826 | 0.923 |
| NetLSD-heat | **0.996** | 0.939 | 0.948 | 0.956 | **0.950** | 0.794 |
| NetLSD-wave | **0.996** | 0.943 | **0.970** | **0.983** | 0.614 | **0.955** |
| RandGIN | 0.938 | 0.947 | 0.132 | 0.527 | 0.067 | 0.285 |

We note that some transitions are easier to detect by all the kernels: for heterogeneity, all kernels show relatively good and stable performance. In contrast, for communities, geometry, dimensionality, and complementarity the difference in performance between different kernels is large: some kernels have correlations larger than 0.97, while others may have near-zero performance, which suggests that these kernels are completely insensitive to these characteristics. Finally, while most kernels perform well on the density transition, this turns out to be challenging for NSPDK. Let us now go over each kernel individually and summarize the results.

**The Degree histogram kernel** is added to our analysis as it is similar to the degree kernel used in previous studies O'Bray et al. (2022); Thompson et al. (2022). As expected, this kernel can perfectly detect the transformations of density and heterogeneity since they directly affect the degree distribution. However, the degree histogram kernel is not sensitive to all other transformations. This shows the limitations of using simple graph characteristics to compare graph distributions and motivates the analysis of more advanced kernels that we conduct in our study.

**The Weisfeiler-Leman kernel** and the related **WL-OA kernel** are considered to be powerful graph kernels. However, in our experiments, we observe that they only perform well on interpolations that are well detected by the degree histogram kernel. Indeed, WL and WL-OA are the best kernels for heterogeneity, which is expected as the first step of the WL procedure is based on node degrees. Similarly, both of them can detect the density transformation well. In contrast, for all the remaining interpolations, the Spearman correlation coefficient is quite small. Based on these results, we can assume that WL-based kernels are mostly sensitive to the degree distributions. However, it is worth noting that both WL and WL-OA dominate the degree histogram kernel for all interpolations since they are based on more advanced graph statistics that go beyond immediate neighbors.

Comparing WL and WL-OA, we note that the latter dominates for all considered transformations. This confirms that the procedure of node matching improves the sensitivity of this kernel.

**The Shortest Path kernel** has quite stable performance. Indeed, its correlation values always exceed 0.9 (in fact, it is the only kernel with this property), even for the most difficult transitions — communities, geometry, dimensionality, complementarity. Interestingly, SP is not among the best kernels for simple transitions (heterogeneity and density), while still having good performance. Notably, SP is the best for dimensionality, which is arguably the most subtle transition. This can be explained by the fact that shortest path lengths capture both local and global information of the graphs, making this kernel sensitive to non-trivial transitions.

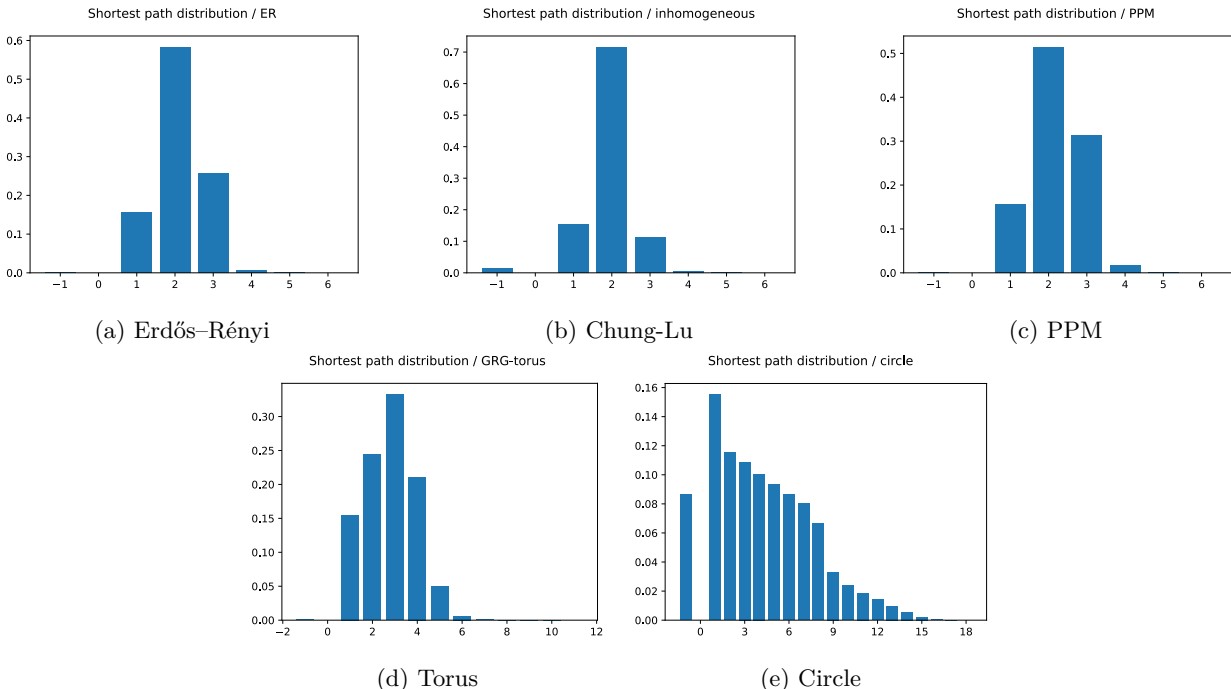

Figure 2: The distribution of shortest path lengths for different graph generators, -1 corresponds to disconnected node pairs.

Figure 2 shows the shortest path distribution for different graph generators. It can be clearly seen that all models are distinguishable. In particular, we see that the SP kernel is sensitive to dimensionality: the Circle results in much longer path lengths than the Torus.

**The Graphlet kernel** is another good option in terms of the overall performance: it is among the best-performing kernels for all interpolations but dimensionality, where the difference with the SP kernel is noticeable. The poorer performance for dimensionality may be explained by the fact that the graphlet kernel is not able to capture the global properties of graphs which are affected when we vary the dimension (e.g., its diameter). Figure 3 additionally illustrates the difficulty of the dimensionality transition: it turns out that for small values of $\theta$ (when the height $h$ is not too small), the distribution of graphlets does not change much, and thus the MMD values are close to zero up to $\theta = 0.7$. We also see that increasing the kernel sizes from 3 to 4 significantly improves the performance for dimensionality since this change makes the considered neighborhoods larger. For all other interpolations, increasing graphlet sizes does not lead to noticeable improvements.

**The Neighborhood Subgraph Pairwise Distance kernel** was previously used in the analysis by Thompson et al. (2022) and has shown reasonably good results in their experiments. In contrast, our analysis with diverse transformations shows certain shortcomings of this kernel. In particular, NSPDK turns out to be insensitive to community structure and density. Other interpolations that are hard to detect for this kernel are dimensionality and complementarity. We hypothesize that such poor performance for some of the transitions can be explained by a particular graph invariant used to compare two rooted subgraphs (as the exact graph isomorphism is infeasible). Unfortunately, this graph invariant also makes this kernel harder to theoretically analyze or intuitively explain.

Let us note that NSPDK has the advantage that it can be used for graphs with node attributes. However, when graphs do not have node labels or attributes, we do not advise using this kernel.

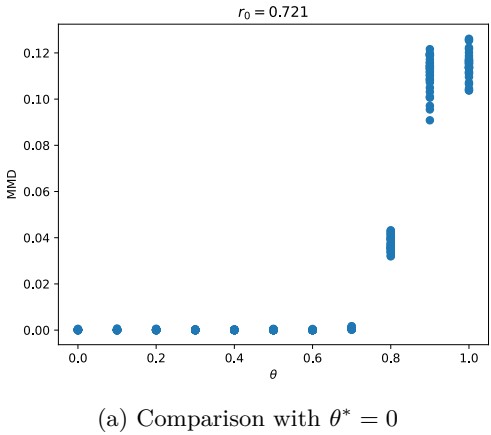 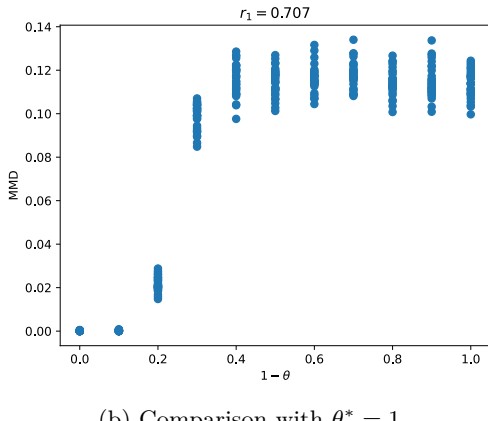

(a) Comparison with $\theta^* = 0$            (b) Comparison with $\theta^* = 1$

Figure 3: Scatter plots of the MMD values of the Graphlet-3 kernel for the torus↔circle interpolation.

**The Pyramid Match kernel** is another kernel that seems to be sensitive to all the characteristics. Similarly to the graphlet kernel, PM does not perform so well on dimensionality. Let us also note that PM is dominated by Graphlet-4 for each interpolation. On the other hand, this kernel is scalable and thus it is a good option for larger graphs where the Graphlet kernel cannot be applied. Similarly to NSPDK, the pyramid match kernel has a complex construction procedure and thus its performance is hard to intuitively interpret. Still, our diverse transformations allow us to get some insights into such complex graph kernels.

**The NetLSD representations** are also sensitive to most of the properties. On almost all the transitions, NetLSD-wave outperforms NetLSD-heat. The notable exception is Dimensionality, which is arguably the most challenging interpolation. Here, NetLSD-heat is the second-best, while the performance of NetLSD-wave is quite poor. We illustrate the difference between NetLSD-heat and NetLSD-wave for this transition in Figure 5, where each cell shows the MMD value for two samples of graphs: corresponding to steps $\theta_1$ and $\theta_2$. As expected, in both cases we see near-zero MMD values on the diagonal $\theta_1 = \theta_2$ since the distributions are the same. For a good kernel, we also expect that MMD increases monotonically as we move away from the diagonal while keeping $\theta_1$ (or $\theta_2$) fixed since the distributions become more different. In particular, for $\theta_2 = 1$, we expect the distance to monotonically increase when we move from $\theta_1 = 1$ to $\theta_1 = 0$. This is indeed the case for NetLSD-heat, but surprisingly not the case for NetLSD-wave.

Figure 5b shows that the distance between graph samples for $\theta_1 = 0.6$ and $\theta_2 = 1$ is larger than the distance between $\theta_1 = 0$ and $\theta_2 = 1$, which is quite unexpected. To additionally illustrate this phenomenon, Figure 4 shows the NetLSD-wave traces from which the graph representations are obtained. One can see that the trace for $\theta = 0$ is indeed closer to $\theta = 1.0$ than $\theta = 0.6$. Taking into account quite complex nature of NetLSD-wave graph representation, we currently cannot explain this behavior. However, our framework allows one to detect such peculiar properties of graph representations.

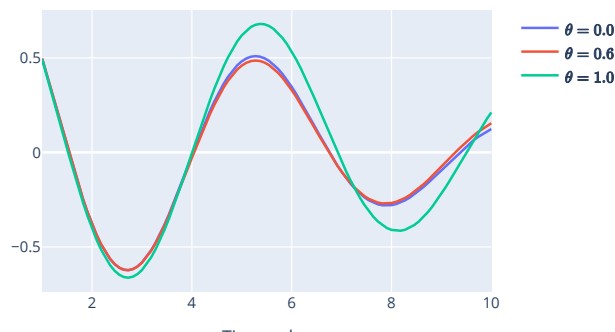

Figure 4: NetLSD-wave traces for Dimensionality.

**Random GIN representations** were proposed in Thompson et al. (2022) in the context of evaluating graph generative models. We note that there are some differences in our setup. First, our graphs do not have node features, while being able to process such features is one of the main advantages of RandGIN. Second, Thompson et al. (2022) use GIN representations to compute measures like precision, recall, or Fréchet Distance. Instead, we use RandGIN to construct a graph kernel that can be used with the MMD framework: we do this to compare all kernels and representations in the unified setup. Our results show that when

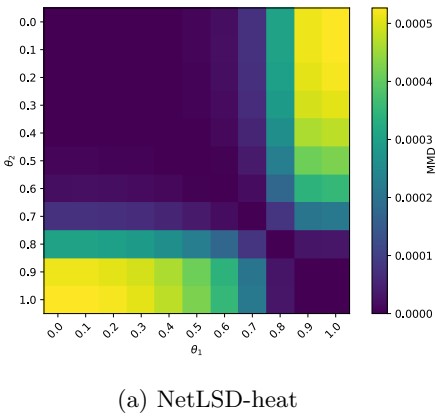

(a) NetLSD-heat

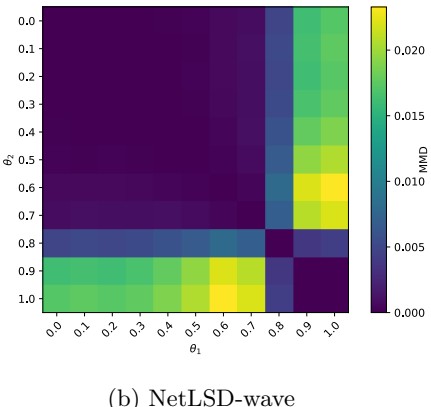

(b) NetLSD-wave

Figure 5: Heatmap of MMD values for Dimensionality transition. MMD measures the distance between two samples of graphs corresponding to steps $\theta_1$ and $\theta_2$.

Table 2: Sensitivity of kernels to various structural characteristics for graphs on 1000 nodes.

|  | Heterogeneity (ER↔CL) | Communities (ER↔PP) | Geometry (ER↔Torus) | Dimensionality (Torus↔Circle) | Complementarity (ER↔SC) |
|---|---|---|---|---|---|
| WL | 0.996 | -0.030 | 0.741 | 0.226 | 0.561 |
| WL-OA | 0.996 | 0.099 | 0.948 | 0.210 | 0.871 |
| PM | 0.975 | 0.931 | 0.989 | 0.238 | 0.981 |
| NetLSD Heat | 0.910 | 0.530 | 0.985 | 0.631 | 0.958 |
| NetLSD Wave | 0.957 | 0.648 | 0.993 | 0.211 | 0.981 |
| RandGIN | 0.931 | -0.041 | 0.849 | 0.219 | -0.080 |
| Degree | 0.996 | -0.034 | -0.009 | -0.015 | 0.132 |

used within our framework, the kernel based on RandGIN is not sensitive to interpolations as communities, geometry, dimensionality, and complementarity. We see that the performance of RandGIN is dominated by the performance of WL-OA. This can be expected since the expressive power of message-passing neural networks is known to be upper-bounded by the WL test (Xu et al., 2019).

## 4.3 Experiments on larger graphs

Results for scalable kernels on larger graph datasets with 1000 nodes are shown in Table 2. To reduce the computation time of these experiments, we set the number of packs to $s = 10$ instead of 30. We see that these results are consistent with those discussed above for smaller graphs. In particular, NetLSD-heat is better than NetLSD-wave for all transitions but Dimensionality, while NetLSD-heat is the best scalable kernel for Dimensionality transition.

## 5 Conclusion

In this paper, we analyze the expressivity of graph kernels in the context of evaluating the performance of graph generative models. We first note that previous research on this subject used only a limited set of graph kernels. We significantly extend this set and consider a list of long-known graph kernels. To conduct a detailed analysis of the kernels, we develop a framework that allows us to check whether a kernel is insensitive to a particular high-level structural graph property. For this, we carefully design an experimental setup based on interpolations between random graph generators differing in a particular structural property.

The results of our experiments confirm the main points of our paper. First, it is important to consider graph kernels beyond those usually considered for evaluating the performance of graph generative models. Indeed,

some well-known kernels that show good performance in our experiments have been previously overlooked in the literature on this subject. Examples of such kernels include Shortest Path and Pyramid Match kernels. Second, we observe that the wide diversity of the considered structural properties is critical for a thorough evaluation: it is often the case that a particular graph kernel is sensitive to some characteristics while being insensitive to others.

The results shown in Table 1 can be useful when deciding which graph kernel is most suitable when evaluating generative models in a particular application. For example, our experiments show that the popular Weisfeiler-Lehman kernel is insensitive to geometry. In certain applications, geometry can be especially relevant, e.g., for the application of molecular modeling because atoms have spatial locations. In this case, our analysis shows that using the WL kernel is not a good option. Let us also note that kernels with different properties can potentially be combined with each other by, e.g., averaging their values. Indeed, if one kernel is sensitive to a particular characteristic while another kernel is sensitive to another characteristic, then their combination is expected to be sensitive to both of them. However, a detailed analysis of such combinations is beyond the scope of the current paper.

### Limitations

It is important to keep in mind that it can be challenging (or even impossible) to change one graph characteristic keeping *all* other properties fixed. While we attempt to isolate the characteristic of interest, it is likely that some other properties are affected to some extent. However, let us note that if a kernel turns out to be insensitive to a particular interpolation this implies that it is insensitive to the underlying characteristics. Also, for clustering we consider two different ways to increase it, thus reducing the possible bias from a particular generator.

Another limitation of our work is that we only consider graphs without attributes. Our general framework easily extends to attributed graphs, but the main challenge would be to design meaningful graph generators for this scenario: there are many ways to combine standard graph generators with different attribute distributions.

Additionally, our study considers relatively small graphs. Our main motivation is that evaluating graph generative models using graph kernels is typically applied to such small networks. For example, in molecular modeling, the graphs consist of a relatively small number of atoms. Other applications where small graphs are relevant include modeling ego networks in sociology. Additionally, we note that not all kernels are scalable: for instance, the graphlet kernel has a prohibitively high computational cost on large networks. We consider smaller graphs so that we can include all popular graph kernels in our experiments.

Finally, we note that our work is an empirical study of the sensitivity of graph kernels to some structural shifts. Theoretical analysis supporting the obtained conclusions would be beneficial. However, for some kernels, such an analysis would be trivial, e.g., the degree histogram is clearly sensitive to shifts in the degree distribution, but not sensitive to shifts in other characteristics that keep the degree distribution unchanged. On the other hand, for most of the kernels, such an analysis seems intractable due to their complex structures.

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

# A    Derivations for graph generators

## A.1    Random geometric models

For a two-dimensional torus with width 1 and height $h \leq 1$, we find the radius $r$ that leads to $m$ edges in expectation. Two nodes are connected when their distance is smaller than $r$, i.e., whenever the second node is inside a circle with radius $r$ around the first node. If $h \geq 2r$, then the surface area of this circle is simply $\pi r^2$, while the total area of the torus is $h$. Therefore, the expected edge density is $\pi r^2$. Solving $\binom{n}{2} \cdot \frac{\pi}{h} r^2 = m$ leads to

$$r = \sqrt{\frac{h \cdot m}{\pi \cdot \binom{n}{2}}},$$

which is valid as long as $2r \leq h$, i.e., whenever $h \geq \frac{4m}{\pi \cdot \binom{n}{2}}$. Otherwise, the circle overlaps itself, which leads to a smaller surface area, given by the following integral

$$A(h,r) = \int_{-h/2}^{h/2} \int_{-\sqrt{r^2-y^2}}^{\sqrt{r^2-y^2}} 1 dx dy = 2r^2 \int_{-\frac{h}{2r}}^{\frac{h}{2r}} \sqrt{1-u^2} du = r^2 h \sqrt{1 - \left(\frac{h}{2r}\right)^2} + 2r^2 \arcsin\left(\frac{h}{2r}\right).$$

To obtain $m$ edges in expectation, we need to choose $r$ such that $A(h,r)/h = m/\binom{n}{2}$. After substituting $z = \frac{h}{2r}$, we get

$$\frac{m}{\binom{n}{2}} = \frac{h}{2} \left( \sqrt{\frac{1}{z^2} - 1} + \frac{1}{z^2} \arcsin(z) \right) = \frac{h}{2} f(z).$$

We then use Newton-Raphson iteration to solve $f(z) = \frac{2m}{h\binom{n}{2}}$. Let us denote the obtained value by $z = f^{-1}\left(\frac{2m}{h\binom{n}{2}}\right)$. Finally, we take the radius $r = \frac{h}{2z}$. In summary, we choose the radius as

$$r(h,m) = \begin{cases} \sqrt{\frac{h \cdot m}{\pi \cdot \binom{n}{2}}} & \text{if } h \geq \frac{4m}{\pi \cdot \binom{n}{2}}, \\ \dfrac{h}{2f^{-1}\left(\frac{2m}{h\binom{n}{2}}\right)} & \text{else.} \end{cases} \tag{3}$$

|  | Heterogeneity (ER↔CL) | Communities (ER↔PP) | Geometry (ER↔Torus) | Dimensionality (Torus↔Circle) | Complementarity (ER↔SC) |
|---|---|---|---|---|---|
| Degree | 0.14 | 0.15 | 0.13 | 0.14 | 0.14 |
| SP | 1.13 | 1.13 | 1.13 | 1.05 | 1.13 |
| WL | 0.15 | 0.19 | 0.18 | 0.19 | 0.19 |
| WL-OA | 1.16 | 1.27 | 1.11 | 1.03 | 1.18 |
| Graphlet-3 | 11.32 | 8.70 | 7.68 | 5.97 | 9.00 |
| Graphlet-4 | 150.86 | 85.37 | 70.08 | 35.75 | 91.08 |
| NSPDK | 41.37 | 40.46 | 36.70 | 24.83 | 38.48 |
| PM | 0.63 | 0.64 | 0.62 | 0.60 | 0.63 |
| NetLSD-heat | 0.74 | 0.78 | 0.73 | 0.89 | 0.81 |
| NetLSD-wave | 0.88 | 0.89 | 0.85 | 1.04 | 0.96 |
| RandGIN | 0.43 | 0.46 | 0.43 | 0.44 | 0.45 |

Table 3: The average computation times (in seconds) for comparing two sets of $g = 100$ graphs, each consisting of $n = 50$ vertices. The shown times are averaged over the $|\Theta| = 11$ interpolation steps. The experiments were conducted on a laptop with AMD Ryzen 7 8840HS CPU and 16GB RAM.

The one-dimensional torus (the circle) is approximated as $h \downarrow 0$. For a circle, we need $r = \frac{m}{2\binom{n}{2}}$ to obtain $m$ edges in expectation. In the remainder, we show that $f^{-1}\left(\frac{2m}{h\binom{n}{2}}\right) \sim \frac{h}{m}\binom{n}{2}$ as $h \downarrow 0$, so that indeed $r(h,m) \to \frac{m}{2\binom{n}{2}}$: first, note that $f(z) \sim 2/z$ as $z \downarrow 0$, so that

$$f\left(\frac{h}{m}\binom{n}{2}\right) = \frac{2m}{h\binom{n}{2}} + o(h^{-1}) \Rightarrow f^{-1}\left(\frac{2m}{h\binom{n}{2}}\right) = f^{-1}\left(f\left(\frac{h}{m}\binom{n}{2}\right) + o(h^{-1})\right).$$

Next, we take the Taylor expansion of $f^{-1}$ around $f\left(\frac{h}{m}\binom{n}{2}\right)$:

$$f^{-1}\left(f\left(\frac{h}{m}\binom{n}{2}\right) + o(h^{-1})\right) = f^{-1}\left(f\left(\frac{h}{m}\binom{n}{2}\right)\right) + \frac{o(h^{-1})}{f'\left(f^{-1}\left(f\left(\frac{h}{m}\binom{n}{2}\right)\right)\right)} = \frac{h}{m}\binom{n}{2} + \frac{o(h^{-1})}{f'\left(\frac{h}{m}\binom{n}{2}\right)}.$$

Finally, the derivative is given by $f'(z) = -2\frac{\arcsin(z)}{z^3} \sim z^{-2}$, so that the second term is $o(h)$. In conclusion, we have

$$f^{-1}\left(\frac{2m}{h\binom{n}{2}}\right) = \frac{h}{m}\binom{n}{2} + o(h),$$

as required.

### A.2 Planted partition model

For the transition between ER and PP, we want $\theta = 1$ to lead to every node having one connection to the other community in expectation. We use equation 2 with $\lambda = 1 + c(n,m) \cdot \theta$. We compute

$$1 = p_{out}(1 + c(n,m)) \cdot \frac{n}{2} = \frac{2m}{n \cdot (2 + c(n,m)) - 2(1 + c(n,m))}.$$

This leads to the solution

$$c(n,m) = 2 \cdot \frac{m + 1 - n}{n - 2}.$$

## B  Computation times

Table 3 shows the average computation time for comparing two graph sets for each of the kernels and graph transitions. Note that the experiments of Table 1 require $s \cdot |\Theta| = 330$ such comparisons for each transition and kernel.

