# OpenReview forum: "Evaluating Graph Generative Models with Graph Kernels: What Structural Characteristics Are Captured?"
_TMLR — Accepted by TMLR_

### Review · Reviewer_oYiU · 2024-10-11

**Summary Of Contributions:**

The paper addresses the important problem of evaluating graph generative models using graph kernels. It conducts comprehensive benchmarks to analyze the sensitivity of different graph kernels to various high-level structural properties of graphs, such as heterogeneity of degree distribution, community structure, and latent geometry. It's worth noting that this is a benchmark paper and no new models are proposed.

**Audience:**

Yes

**Claims And Evidence:**

Yes

**Requested Changes:**

1. Please state how the graph kernel hyperparameters are chosen, and provide additional results for other hyperparameter settings if feasible.
2. The notations for the numbers of iterations of WL{,-OA} kernels $l$ and MMD values $\ell$ are not obvious to distinguish, consider replacing one of them.

**Strengths And Weaknesses:**

Pros:
1. Graph kernels are widely adopted in graph-level tasks, but systematic studies of the properties of different kinds of graph kernels were lacking. The paper fills the gap and contributes to the understanding of which graph kernels are suitable for evaluating generative models by highlighting the strengths and weaknesses of different kernels in capturing specific graph properties.
2. The design of the evaluation framework appears to be reasonable and original.
3. The benchmark results and the corresponding analyses might be valuable for researchers on graph-level tasks.
4. The overall structure is clear and well-organized.

Cons:
1. The evaluation and proposed interpolation method are based on synthetic graphs. While this is perfectly fine for controlled experiments, it would be more beneficial to see how the proposed framework and insights generalize to real-world datasets such as OGB.
2. Some of the graph kernels have hyperparameters (such as $l$ of the WL kernel) and are described in Section 4.1, but the reason why they are chosen remains unstated. Besides, it's unclear if the benchmark results generalize to other hyperparameter settings.
3. The result discussions are mostly qualitative and lack theoretical guarantees.

---

> ### Author Response · Authors · 2024-11-08
>
> Thank you for your review and feedback! We address the concerns and questions below. We plan to update the paper by incorporating the requested changes before the end of the discussion period.
>
> **Synthetic vs real-world datasets**
>
> In our study, we use synthetic datasets since real-world ones do not allow us to gradually increase or decrease characteristics, while we aim to conduct a detailed analysis of sensitivity to various structural shifts. Experiments based on real-world datasets have been conducted by O’Bray et al. (2022) and Thompson et al. (2022). However, the transformations considered in these works do not allow for isolating specific graph characteristics and thus conducting a similar analysis to our study.
>
> **Choice of hyperparameters**
>
> For the considered kernels, we mostly use their default hyperparameters. This means that our parameters are either the default parameters of the implementation that we use or are the most commonly used parameters in the literature. In some cases, however, the parameter choice is limited by computational complexity of a kernel. For instance, for a graphlet kernel, we cannot consider $k>4$.
>
> Since our research compares graph kernels as ingredients of evaluation measures, we have to fix a particular parameter before running the experiments. Thus, if we want to check the effect of a particular parameter, we need to consider all parameter variants as different kernels. We do this for the graphlet kernel by considering Graphlet-3 and Graphlet-4 as examples. By comparing these two variants, the following observations can be made. First, increasing $k$ for the Graphlet kernel increases its sensitivity, as expected (by the price of computational efficiency). We expect the same to hold for other parameters and kernels, e.g., $l$ in the WL kernel. Second, the performance of Graphlet-4 and Graphlet-3 is similar. For instance, both of them work remarkably well on the Complementarity transition. Thus, we expect similar results for other kernels.
>
> **Theoretical guarantees**
>
> We agree that it would be good to have theoretical results explaining why certain kernels are sensitive to certain characteristics. However, the problem is that for some kernels, such an analysis would be trivial, e.g., the degree histogram is clearly sensitive to shifts in the degree distribution, but not sensitive to shifts in other characteristics that keep the degree distribution unchanged. On the other hand, for most of the kernels such an analysis seems intractable due to their complex structures.

---

### Review · Reviewer_f1sq · 2024-10-27

**Summary Of Contributions:**

This paper focuses on the properties of different graph kernels used for measuring graph similarity. The study empirically assesses the effectiveness of various graph kernels in evaluating graph generative models, with a particular emphasis on their sensitivity to key structural characteristics such as degree distribution, community structure, and latent geometry. The authors highlight the limited number of graph kernels previously considered for model evaluation and propose a framework that includes a broader range of established kernels. They assess which kernels best capture specific structural features by conducting experiments with continuous transitions between random graph models. The findings indicate that some kernels, such as the Shortest Path and Pyramid Match kernels, perform well but are underutilized in current research. This study emphasizes the importance of selecting appropriate kernels that correspond to specific structural properties for more effective model evaluation, providing valuable insights for practitioners.

**Audience:**

Yes

**Broader Impact Concerns:**

I do not have any ethical concerns regarding this paper.

**Claims And Evidence:**

Yes

**Requested Changes:**

1. **Addition of a Deterministic Solution:** I suggest the authors include at least one additional section that proposes a deterministic solution based on the empirical findings presented in the current experiments section. This section could summarize key insights and offer concrete guidance for practitioners. Additionally, I recommend including another section that empirically justifies the proposed deterministic solution in practical scenarios, with respect to both performance and computational efficiency.

2. **Presentation Improvements:** As highlighted in the weaknesses section, certain modifications to the presentation would enhance clarity. In particular, refining the transitions between sections and providing clearer definitions  would improve the overall readability and flow of the paper.

**Strengths And Weaknesses:**

**Overall Evaluation:**

This paper presents a solid contribution to the study of graph kernels for measuring graph similarity, offering several strengths but also some limitations.

**Strengths:**

1. **Systematic Approach:** The paper provides a comprehensive study on the effectiveness and sensitivity of various graph kernels with respect to key structural characteristics. The unified framework proposed helps to ensure fairness in evaluation, which is valuable for consistent comparisons.

2. **Practical Insights:** The findings offer valuable insights for practitioners by highlighting the impact of graph kernel selection on model evaluation outcomes. This provides practical guidance for choosing appropriate kernels based on specific structural characteristics relevant to their use cases.

**Weaknesses:**

1. **Lack of Deterministic Solutions:** While the paper presents interesting findings, it lacks concrete recommendations for practitioners on how to measure graph similarity in real-world applications. For example, if practitioners are looking for a direct method, should they consider combining multiple kernels into an ensemble? Furthermore, the paper does not address the practicality of these kernels in terms of computational efficiency.

2. **Applicability to Node Features:** It is unclear whether the findings and framework extend to graphs with node features (e.g., vector or text features). While this limitation seems inherent to graph kernels rather than the study itself, it does restrict the practical applicability of the results in real-world scenarios where node features are present.

**Presentation Comments:**

1. **Abstract:** In the abstract, the phrase "using such graph modifications" is unclear, as the term "modification" is not well-defined earlier in the text. A clearer explanation is needed to enhance reader understanding.

2. **Introduction:** The transition between the second and third paragraphs feels abrupt. The discussion moves from explaining why graph kernels are proposed for evaluating graph similarity to evaluating graph generative models without sufficient flow. A smoother transition or clearer connection between these topics would improve the readability of the introduction.

---

> ### Author Response · Authors · 2024-11-08
>
> Thank you for your review and feedback! We address the concerns and questions below.
>
> **Presentation Comments**
>
> Thank you for your suggestions, we’ll update the paper accordingly before the end of the discussion period.
>
> **Deterministic Solution**
>
> Providing deterministic recommendations is highly non-trivial since the preferred graph kernel significantly depends on the application. Instead, we provide a framework that allows for a detailed analysis of the sensitivity of kernels and can be applied not only to the kernels considered in the paper but to some other ones that a practitioner finds suitable for an application at hand or that can be developed in future works. We provide some examples of how one can reason about what kernel is better in a particular scenario in the second paragraph on page 2 and in the last paragraph of Section 5. Your suggestion about combining several kernels into an ensemble to improve the sensitivity is reasonable, we will discuss this approach in the revised paper.
>
> **Computational Efficiency**
>
> Since we use well-established graph kernels, we do not conduct theoretical analysis of their computational complexity in this paper. In the revised version of the paper, we plan to empirically compare the efficiency of kernels used in our study.
>
> **Applicability to Node Features**
>
> In this work, we focus on the sensitivity of graph kernels to graph characteristics. To extend the results to node features, we would have to model the dependencies between node features and the graph characteristics, which is highly non-trivial since we would have to make several assumptions about the feature distribution and the interplay between features and network structure. This would significantly complicate the analysis. Moreover, different types of kernels allow for different node attributes, which makes it impossible to compare kernels that require a certain type of attributes (e.g. labels) to kernels that require other attributes (e.g. real-valued attributes). This is the reason that we focus on a comparison based on network topology alone. We think it is useful to first understand kernels for unlabeled graphs before studying kernels for labeled graphs, which we leave for future research.
>
> Let us also note that if a kernel is sensitive to a particular structural characteristic, then it is expected to still be sensitive to it when features are added: adding features usually makes it easier to distinguish graphs. On the other hand, if a kernel is insensitive to a particular structural characteristic, then it is still insensitive to it when the node features do not carry information about the given graph characteristic (e.g., features are not community labels, degrees, triangle counts, etc). Thus, even for the labeled case, it is natural to require a kernel to be sensitive to structural characteristics of interest.

---

### Review · Reviewer_9owH · 2024-10-29

**Summary Of Contributions:**

The authors investigate the sensitivity of various Graph Kernels, employed in graph learning, for graph generative model evaluation tasks. They examine diverse graph characteristics and generate Erdős-Rényi graphs as reference points, alongside alternative random graph methods that vary in properties such as density, degree heterogeneity, and clustering. These random graph families are chosen such that their characteristics are controllable via a parameter. The sensitivity of these methods is assessed by computing the correlation coefficients between this parameter and the Maximum Mean Discrepancy (MMD) distance derived using these kernels. Experiments are conducted on graphs without node or edge features, with two sets of graphs differing in size: one set of smaller graphs and another set of larger graphs with 1000 nodes.

**Audience:**

Yes

**Broader Impact Concerns:**

This does not seem to be applicable to this work.

**Claims And Evidence:**

Yes

**Requested Changes:**

1. Computational Analysis: Please include an analysis of both the theoretical time complexity and wall-clock computation time of the graph kernels. Addressing at least one of these aspects is necessary, including both is recommended for a comprehensive evaluation.

2. Comparison with Graph Laplacian Metric: Please compare the presented methods with the graph Laplacian metric. This comparison is essential for a thorough assessment of the methods' performance.

3. Formatting Improvement: The numbers and texts in the figures are too small, and the quality of the figures is low when zoomed in. Please correct these figures. Additionally, the figures on page 11 have excess space, and the formatting could be improved for better presentation.

**Strengths And Weaknesses:**

## Strengths

1. Relevancy: The problem addressed is relevant to the community and is considered an important issue. There is no clear solution regarding which metric is proper, and this work takes a step in a direction that helps to clarify this problem.
2. Accessible and adequate descriptions: The authors provide clear and comprehensible descriptions of the graph kernels and random graph generation methods used in the study.
3. Selection of methods: The choice of graph kernels and random graph generators appears well-reasoned and appropriate for the research questions addressed.
4. Thorough analysis: The analysis covers major characteristics of the graphs considered in the context of learning on graphs, with experiments on all these aspects.
5. Clarity: The paper is well-written, with a clear structure and coherent argumentation, making it easy for the reader to follow the authors' line of thought.

## Weaknesses

1. Lack of theoretical analysis: The paper does not present or discuss any theory related to these topics. What is the theoretical expressivity of each of these kernels? Why or why not should these kernels be sensitive to some characteristics of the graph datasets?

2. Absence of computational analysis: There is no analysis of the time complexity or experimental computation time of the graph kernels considered. Generally, there is a trade-off between the speed of calculating the metric and its effectiveness. For instance, with infinite time, we could exactly calculate the edit distances of the graphs, providing a very accurate metric.

3. Limited graph size: The experiments focus on graph datasets that are smaller than many real-world graphs. Both the graph sizes and the number of samples are very small, and since the graphs are synthetic, the analysis could be adapted for any size.

4. Insufficient sampling: The use of only 30 sets of graphs and 100 MMD samples may not provide a sufficiently robust statistical basis for the conclusions drawn.

5. Absence of node and edge features: Real-world graphs often have node and edge features, which can be crucial for many applications. However, the methods presented focus solely on graph structure.

6. Lack of real-world data analysis: The paper does not include any analysis of real-world datasets, which could provide valuable insights into the practical applicability of the methods.

7. Limited comparison with classical metrics: Apart from node degree, the paper does not compare the presented methods with other classical metrics, such as clustering coefficient vectors or graph Laplacian eigenvalues introduced in [1].

8. Omission of sensitivity analysis: The paper does not include a sensitivity analysis, which has been shown to be crucial and to provide important insights into the robustness of the methods, similar to the analysis conducted in [2].


[1] Liao, Renjie, et al. "Efficient graph generation with graph recurrent attention networks." Advances in neural information processing systems 32 (2019).

[2] Leslie O’Bray, Max Horn, Bastian Rieck, and Karsten Borgwardt. Evaluation metrics for graph generative
models: Problems, pitfalls, and practical solutions. In International Conference on Learning Representations, 2022.

---

> ### Author Response · Authors · 2024-11-08
> **Response [Part 1/2]**
>
> Thank you for the detailed feedback! We address the concerns and questions below.
>
> **Lack of theoretical analysis**
>
> We agree that it would be good to have theoretical results explaining why certain kernels are sensitive to certain characteristics. However, the problem is that for some kernels, such an analysis would be trivial, e.g., the degree histogram is clearly sensitive to shifts in the degree distribution, but not sensitive to shifts in other characteristics that keep the degree distribution unchanged. On the other hand, for most of the kernels such an analysis seems intractable due to their complex structures.
>
> **Computational analysis**
>
> Since we use well-established graph kernels, we do not conduct a theoretical analysis of their computational complexity in this paper. What we plan to do in the revised version of the paper is to measure the running time of the kernels for the graphs considered in this paper. Then, the overall complexity can be obtained by multiplying this value by the number of kernel computations $l\cdot|\Theta|\cdot(2 g)^2$. We hope this experiment would address the raised concern.
>
> **Limited graph size**
>
> Let us note that in Section 4.3, we conduct the experiments for graphs consisting of 1000 nodes. The results (Table 2) are consistent with the results for small graphs, which suggests that the conclusions would not change if we further increase the graphs' sizes. Also, note that our comparative analysis of graph kernels requires much more computational resources than one would need to apply these kernels in practice: we consider several graph kernels, several structural characteristics, and for each characteristic we have several steps, and for each step we generate several sets of graphs. This makes our experiments computationally heavy. Because of this, we decided to limit our experiments to graphs with size 50 or (for some kernels) 1000.
>
> **Insufficient sampling**
>
> Note that we consider 30 sets of 100 graphs for each of the 11 transition steps. Each correlation value is computed based on $11\cdot 30=330$ MMD values, each of which is calculated based on two sets of 100 graphs each.
> One could run these experiments for larger values of these parameters, but the total running time of the experiments (several days) is already quite significant. We also noticed that the experimental results do not differ much between different runs of the same experiment.
>
> **Absence of node and edge features**
>
> In this work, we focus on the sensitivity of graph kernels to graph characteristics. To extend the results to node features, we would have to model the dependencies between node features and the graph characteristics, which is highly non-trivial since we would have to make several assumptions about the feature distribution and the interplay between features and network structure. This would significantly complicate the analysis. Moreover, different types of kernels allow for different node attributes, which makes it impossible to compare kernels that require a certain type of attributes (e.g. labels) to kernels that require other attributes (e.g. real-valued attributes). This is the reason that we focus on a comparison based on network topology alone. We think it is useful to first understand kernels for unlabeled graphs before studying kernels for labeled graphs, which we leave for future research.
>
> Let us also note that if a kernel is sensitive to a particular structural characteristic, then it is expected to still be sensitive to it when features are added: adding features usually makes it easier to distinguish graphs. On the other hand, if a kernel is insensitive to a particular structural characteristic, then it is still insensitive to it when the node features do not carry information about the given graph characteristic (e.g., features are not community labels, degrees, triangle counts, etc). Thus, even for the labeled case, it is natural to require a kernel to be sensitive to structural characteristics of interest.

---

> ### Author Response · Authors · 2024-11-08
> **Response [Part 2/2]**
>
> **Lack of real-world data analysis**
>
> In our study, we use synthetic datasets since real-world ones do not allow us to gradually increase or decrease characteristics, while we aim to conduct a detailed analysis of sensitivity to various structural shifts. Experiments based on real-world datasets have been conducted by O’Bray et al. (2022) and Thompson et al. (2022). However, the transformations considered in these works do not allow for isolating specific graph characteristics and thus conducting a similar analysis to our study.
>
> **Limited comparison with classical metrics**
>
> Firstly, note that the clustering coefficient vector and Laplacian eigenvalues are graph representations and not graph kernels. Therefore, we would have to transform them into kernels using, e.g., RBF (which requires an additional parameter) or an inner product. While this is often done when comparing graph generative models, we argue that using well-established kernels directly is more straightforward. We discuss this in the last paragraph on page 4.
>
> Additionally, we note that Graphlet3 counts the number of triangles and open wedges, which is similar to computing the clustering coefficient. Also, the NetLSD kernel is based on the graph Laplacian spectrum.
>
> **Omission of sensitivity analysis**
>
> Could you please clarify what type of sensitivity analysis you are referring to in this comment?
>
> **Formatting Improvement**
>
> Thank you for the suggestions, we will improve the formatting and update the paper before the end of the discussion period.

---

### Author Response · Authors · 2024-11-12
**Updated paper**

We would like to thank the reviewers for their constructive feedback. Following their suggestions, we’ve updated the paper.

The following modifications have been made:

- Computational analysis is added in Appendix B (Reviews 9owH and f1sq)
- Quality of figures is improved (Review 9owH)
- Text improvements in abstract and introduction (Review f1sq)
- Added a comment about combining different kernels to Conclusion (Review f1sq)
- Discuss the choice of the hyperparameters in Section 4.1 (Review oYiU)
- Changed ‘\ell’ to ‘s’ (Review oYiU)

---

### Decision · Action_Editor_XjSy · 2024-12-05

**Recommendation:** Accept with minor revision

**Comment:**

I suggest following suggestions by the Reviewers and updating the manuscript:

- Explain the rationale for the evaluation using synthesis data and discuss the evaluation on real-world graph data as a limitation or future work.
- Discuss theoretical guarantees (e.g., representation capabilities of graph kernels) as a limitation or future work.

We ask the authors to add **Statement of Broader Impact.** (see [Author guidelines](https://jmlr.org/tmlr/author-guide.html)). Although the reviewers did not point out any broader impact concern, I suggest discuss it in the manuscript.

I suggest fixing the following minor typos:
- *Weisfeiler-Lehman optimal assignment (WL-OA) kernel was introduced by (Kriege et al., 2016)* -> Kriege et al. (2016)
- a dynamic system → dynamical system

**Audience:**

Many graph kernels have been proposed in graph machine learning research, and it is an important research question: Which graph kernels are compatible with what kind of graph structures? In addition, evaluation of graph generative models is an important issue. For example, in graph neural network (GNN) research, graph generative models using GNNs are one of the active topics, and appropriate evaluation methods for these models are considered an important issue. Therefore, this paper, which proposes a framework for evaluating graph generative models, is of interest to the TMLR audience.

**Claims And Evidence:**

This paper proposes a framework for evaluating graph generative models to investigate the sensitivity of graph kernels to graph structures. The claims of this paper are the following:

- Existing research under-explores this topic, as it has only been done for a limited number of kernels. On the other hand, the proposed framework supports a wider range of graph kernels with various graph structures.
- Many graph kernels can capture specific structures sensitively but not others. Among others, the shortest path kernel, which is not often used for graph generative model evaluation, performs best in this study.
- The proposed framework allows practitioners to choose the best graph kernel for their application.

One reviewer pointed out that there is no specific recommendation on which graph kernel to use in the practical scenario, which makes it difficult for practitioners to use. In response, the authors argued that it is difficult to give a deterministic solution and made some suggestions to practitioners, which looks reasonable to me. See the discussion [here](https://openreview.net/forum?id=d9MhajJT04&noteId=JYU1GYLnWJ) for details.

The reviewers pointed out the limitations of the paper in the following aspects:

- The numerical experiments do not employ large graph data.
- Discussions on computational complexity are lacking.
- The numerical experiments employ synthesis data only.
- The proposed framework does not support graphs with node or edge features.
- Theoretical guarantees are lacking.

The revised paper supports the first two aspects. For the third and fourth aspects, the authors adequately explained the rationale for designing the numerical experiments. See the discussions for details ([1](https://openreview.net/forum?id=d9MhajJT04) and [2](https://openreview.net/forum?id=d9MhajJT04)). The final point is not addressed in the paper nor responses, if I get all the information. The argument is stronger if we have theoretical guarantees. Nevertheless, the claim is still valid within the numerical experiment settings, especially in Table 1.

In conclusion, the above claims are considered adequately supported by appropriate evidence.